# Non-Specific Elevated Serum Free Fatty Acids in Lung Cancer Patients: Nutritional or Pathological?

**DOI:** 10.3390/nu16172884

**Published:** 2024-08-28

**Authors:** Yelin Shao, Sicong Wang, Xiaohang Xu, Ce Sun, Fei Cai, Qian Guo, Ming Wu, Min Yang, Xifeng Wu

**Affiliations:** 1Department of Big Data in Health Science, School of Public Health, Center of Clinical Big Data and Analytics of The Second Affiliated Hospital, Zhejiang University School of Medicine, Hangzhou 310058, China; 12118623@zju.edu.cn (Y.S.); wangsicong@zju.edu.cn (S.W.); 12018344@zju.edu.cn (X.X.); 22218838@zju.edu.cn (C.S.); 22218258@zju.edu.cn (F.C.); 22118920@zju.edu.cn (Q.G.); 2National Institute for Data Science in Health and Medicine, Zhejiang University, Hangzhou 310058, China; 3Department of Thoracic Surgery, Second Affiliated Hospital, Zhejiang University School of Medicine, Hangzhou 310058, China; iwuming22@zju.edu.cn; 4The Key Laboratory of Intelligent Preventive Medicine of Zhejiang Province, Hangzhou 310058, China; 5School of Medicine and Health Science, George Washington University, Washington, DC 20052, USA

**Keywords:** lung cancer, serum free fatty acids, nutrition, tumor metabolism, immunity, LC-MS/MS

## Abstract

Importance: The reprogramming of lipid metabolism is a significant feature of tumors, yet the circulating levels of fatty acids in lung cancer patients remain to be explored. Moreover, the association between fatty acid levels and related factors, including nutritional intake, tumor metabolism, and tumor immunity, has been rarely discussed. Objectives: To explore the differences in serum free fatty acids between lung cancer patients and healthy controls, and investigate the factors associated with this phenomenon. Design and participants: A case-control study enrolled 430 primary lung cancer patients and 430 healthy controls. The whole population had a medium [Q1, Q3] age of 48.0 [37.0, 58.9] years, with females comprising 56% of the participants. The absolute quantification of 27 serum free fatty acids (FFAs) was measured using a liquid chromatography–mass spectrometry (LC-MS/MS) detection. Data, including dietary intake, blood indicators, and gene expression of lung tissues, were obtained from questionnaires, blood tests, and RNA-sequencing. Statistical differences in FFA levels between lung cancer patients and healthy controls were investigated, and related contributing factors were explored. Results: Levels of 22 FFAs were significantly higher in lung cancer patients compared to those in healthy controls, with fold changes ranging from 1.14 to 1.69. Lung cancer diagnosis models built with clinical and FFA features yielded an area under the receiver operating characteristic curve (AUROC) of 0.830 (0.780–0.880). Total fatty acids (TFAs), monounsaturated fatty acids (MUFAs), and polyunsaturated fatty acids (PUFAs) showed no significant dietary–serum associations, indicating that the elevations might not be attributed to an excessive intake of relevant fatty acids from the diet. For RNA-sequencing of lung tissues, among the 68 lipid metabolism genes, 26 genes showed significant upregulation (FDR < 0.05), while 33 genes exhibited significant downregulation, indicating the involvement of the fatty acids in the tumor metabolism. Through joint analysis with immune cells and inflammatory factors in the blood, fatty acids might exert suppressing effects on tumor immunity. Conclusions: Lung cancer patients had elevated levels of serum free fatty acids compared to healthy individuals. The elevations might not be attributed to an excessive intake of relevant fatty acids from the diet but related to pathological factors of tumor metabolism and immunity. These findings will complement research on fatty acid metabolism of lung cancer and provide insights into potential intervention targets.

## 1. Introduction

Among all cancer types, lung cancer ranks second in terms of incidence and carries the highest mortality rate worldwide [1]. As a respiratory cancer, lung cancer has been predominantly associated with various factors that impact the respiratory system, including cigarette smoking and air pollution [2]. The metabolic heterogeneity of cancer has attracted extensive attention due to its revelations in pathological mechanisms, potential therapeutic targets, and diagnostic biomarkers [3,4]. Simultaneously, lung cancer screening and early diagnosis have crucial performance in reducing lung cancer mortality. For instance, low-dose computed tomography (LDCT) screening has been shown to decrease mortality rates associated with lung cancer [5], and molecular biomarkers, including autoantibodies, miRNAs, DNA methylation, blood protein profiling, and circulating tumor DNA, were also progressing [6]. They provide a reference for metabolites that also hold potential for non-invasive diagnosis. Therefore, the insight into lung cancer metabolism and discovery of blood-based biomarkers possess potential value both in research and application prospects.

Analogous to other malignancies, lipid metabolism was a pivotal component in the lung cancer metabolic reprogramming [7]. In tumor tissues, the heightened demands for lipids stemmed from accelerated energy metabolism and amplified cell membrane synthesis, involving molecular entities, such as fatty acids, triglycerides, and cholesterol. A pooled analysis comprising 1,445,850 participants revealed that dietary saturated fatty acid intake was a risk factor for lung cancer, whereas polyunsaturated fatty acids acted as protective factors [8]. A meta-analysis of cohort studies including 1,832,880 participants indicated that the blood levels of triglycerides, total cholesterol, and high-density lipoprotein cholesterol all exerted an impact on the lung cancer occurrence [9]. Furthermore, statin medications, which are used to manage lipid abnormalities, have demonstrated enhancements in the survival outcomes for patients with lung cancer [10]. Based on these findings, further exploration into a broader lipid profile is suggested, to depict lipid metabolism in lung cancer and delve deeper into underlying mechanisms.

Free fatty acids (FFAs) present in the bloodstream as small molecules, distinct from triglycerides and phospholipids. Although FFAs make up only about 10% of the total fatty acid content, their roles have been discovered to include providing energy for skeletal muscles and the heart, serving as precursors for signaling molecules, and more [11]. Diseases characterized by free fatty acid abnormalities currently include diabetes, heart disease, kidney disease, schizophrenia, and cancer [12,13,14,15,16,17]. There has been research on total FFAs, as well as targeting pathways of arachidonic acid and linoleic acid [13,18], which reported higher levels of FFAs in lung cancer patients. These studies mainly considered fatty acids’ involvement in tumor metabolism, especially through specific pathways. However, limited information is available about the risk of lung cancer from a broad profile of serum FFAs in relation to dietary nutrition and inflammation perspectives [19,20].

Therefore, we conducted a case-control study to investigate whether serum free fatty acid levels were different between lung cancer (LC) patients and healthy controls (HC), and to evaluate whether the differences were attributable to nutritional or pathological factors.

## 2. Materials and Methods

### 2.1. Study Population

Between 1 August 2020 and 1 August 2023, a total of 430 patients diagnosed with primary lung cancer were recruited at the Second Affiliated Hospital, Zhejiang University School of Medicine in Hangzhou, China. Another 430 healthy individuals who participated in an annual health check-up at the same hospital were also recruited as healthy controls. The 1:1 matching criterion included the same gender, an age difference of within 5 years, and recruitment times within 6 months. All participants provided informed consent for the research. Samples, questionnaires, and blood tests for all participants were conducted following standardized procedures.

### 2.2. Sample and Data Collection

#### 2.2.1. Serum Sample Collection

All participants underwent blood collection under fasting conditions. Venous blood was centrifuged at 1500× *g*, 4 °C, for 10 min, then the serum was obtained into clot activator tubes and immediately stored at −80 °C.

#### 2.2.2. Serum Free Fatty Acid (FFA) Measurement

The quantification of 27 serum FFA concentrations was performed using a liquid chromatography–tandem mass spectrometry (LC-MS/MS) detection. The details of the analytical method are provided in the Appendix A.

Subsequently, 7 subclasses of FFAs, including total fatty acids (TFAs), saturated fatty acids (SFAs), monounsaturated fatty acids (MUFAs), polyunsaturated fatty acids (PUFAs), ω3 polyunsaturated fatty acids (ω3 PUFAs), ω6 polyunsaturated fatty acids (ω6 PUFAs), and ω6/ω3 PUFAs were obtained through calculations [21]. TFAs included all FFAs. SFAs included C12:0, C14:0, C15:0, C17:0, C20:0, and C22:0. MUFAs included C12:1, C14:1, C15:1, C16:1, C17:1, C18:1, C18:1 T, C20:1, and C22:1. PUFAs included C18:2, C18:2 T, C18:3 α, C18:3 γ, C20:2, C20:3, C20:4, C20:5, C22:4, C22:5 ω3, C22:5 ω6, and C22:6 ω3. ω3 PUFAs included C18:3 α, C20:5, C22:5 ω3, and C22:6 ω3. ω6 PUFAs included C18:2, C18:3 γ, C20:3, C20:4, C22:4, and C22:5 ω6.

### 2.3. Clinical Examination and Data Collection

Clinical examinations were performed according to the hospital’s uniform standards. Triglycerides (TG), total cholesterol (TC), low-density lipoprotein cholesterol (LDL-C), high-density lipoprotein cholesterol (HDL-C), apolipoprotein A1 (APOA1), apolipoprotein B (APOB), and C-reactive protein (CRP) were quantified using an automatic biochemical analyzer (Beckman Coulter AU5800, Beckman Corporation, Indianapolis, IN, USA). Neutrophil counts, lymphocyte counts, and leukocyte counts were measured with an automatic blood cell analyzer (Mindray BC-6800, Mindray Corporation, Shenzhen, China). Neuron-specific enolase (NSE), carbohydrate antigen 199 (CA199), carbohydrate antigen 125 (CA125), and carcinoembryonic antigen (CEA) were measured using an automated chemiluminescence immunoassay analyzer (Tellgen TESMI i200, Tellgen Corporation, Shanghai, China).

### 2.4. Questionnaire and Dietary Data Collection

All participants underwent face-to-face interviews conducted by investigators and completed a basic questionnaire to obtain information, including age, gender, body mass index, and smoking status. Among them, 95 participants (90 healthy controls and 5 patients) completed a validated food frequency questionnaire (FFQ). The FFQ included intake frequency and absolute quantification for 63 food items (Appendix A).

Participants’ dietary intake of total energy, monounsaturated fatty acids, polyunsaturated fatty acids, saturated fatty acid, and total fatty acids were calculated based on the China Food Composition (Book 1, 2nd Edition). The individual daily nutrient intake calculation was described previously [22]. Briefly, we initially ordered the food items within each food group from highest to lowest intake using 24 h data [23]. Subsequently, we created the converted food composition table for each group by choosing the food items that made up 80% of the total intake and weighting them according to their constituent ratios.

### 2.5. Diagnostic Prediction Models Construction

The participants were randomly stratified, sampling into a discovery dataset (*n* = 602) and a test set (*n* = 258) by 7:3. Subsequently, we applied the Least Absolute Shrinkage and Selection Operator (LASSO) regression to the discovery dataset to identify a smaller set of features with nonzero coefficients capable of distinguishing LC patients. Logistic regression models were trained with the 19 FFAs (selected from 27 FFAs) and clinical features in the discovery dataset. Following this, the diagnostic model was utilized on the test set. DeLong’s test was performed to investigate the differences in AUROC between the models.

LASSO regression modeling was performed via the glmnet package in R 4.1.2 software.

### 2.6. Selection of Genes in Fatty Acid Metabolism Pathway

Based on pathways related to fatty acid metabolism (hsa01212: fatty acid metabolism; hsa01040: biosynthesis of unsaturated fatty acids; hsa00071: fatty acid degradation; hsa00062: fatty acid elongation; hsa00061: fatty acid biosynthesis) from the KEGG (Kyoto Encyclopedia of Genes and Genomes), we identified 78 genes regulating fatty acid metabolism (Appendix A).

### 2.7. RNA-Sequencing of Lung Tumors and Adjacent Normal Tissues

A total of 346 lung tumor and 401 adjacent non-tumor tissue samples from the LC patients were collected and processed according to the details outlined in the Appendix A. Gene expression levels were quantified using RSEM and batch corrected (Combat algorithm, SVA package, RSEM v0.6). Differential expression genes between lung tumors and adjacent normal tissues were evaluated using R package edgeR.

### 2.8. Statistical Analysis

All data analyses were performed using R 4.1.2 software (Stanford University, Stanford, CA, USA). First, the Shapiro–Wilk normality test was conducted to assess the normality of the data. Then, to investigate differences in FFA levels between the LC and HC groups, Student’s *t* test was employed for normally distributed data, whereas the Mann–Whitney *U* test for skewed distributed data. Multivariate logistic regression analysis was applied to assess FFAs’ effects, as well as interaction effects between FFAs and blood chemistry indicators on lung cancer, adjusting for covariates. Spearman’s rank correlation analysis was utilized to assess the correlations between dietary fatty acids’ intake and serum FFAs, as well as correlations between the blood indicators and serum FFAs. The Benjamini–Hochberg method was applied to control the false discovery rate (FDR).

Subgroup analysis was conducted to examine the distinctions between the LC and HC groups in FFA levels, stratified by age, gender, smoking status, and BMI, respectively.

## 3. Results

### 3.1. Characteristics of the Study Population

As shown in Table 1, the study population had a medium [Q1, Q3] age of 48.0 [37.0, 58.9] years, with females comprising 56.0% of the participants. There were no statistically significant differences in age, gender, body mass index, TG, TC, LDL-C, HDL-C, ApoA1, and ApoB between the lung cancer (LC) group and the healthy control (HC) group. Among study participants, smoking prevalence was 26.0%, in which the LC group had a relatively higher proportion of smokers at 28.6%. The LC group exhibited significantly higher levels of C-reactive protein and neutrophil counts, but lower levels of lymphocyte counts.

In the LC group, in terms of tumor pathological subtypes, adenocarcinoma accounted for 94.4%. In terms of staging, early-stage lung cancer (I–II) accounted for 81.2%, with other cases including in situ carcinoma and advanced-stage lung cancer (III–IV).

### 3.2. Serum Levels of Free Fatty Acids

Table 2 shows the serum fatty acids’ concentrations between the LC group and the HC group. Serum levels of five FFAs, including C15:1, C18:1 T, C18:2 T, C20:0, and C22:0, were not detectable under the LC-MS/MS analytical method and thus were excluded from subsequent analyses. Levels of the other 22 FFAs were all significantly higher in the LC group (fold change > 1, *p* < 0.05) than those in the HC group, with changes ranging from 14% to 70%. Additionally, the calculated levels of certain types of fatty acids, such as MUFAs and PUFAs, were all found to elevate in the lung cancer patients. As shown in Table 2, after adjusting for age and gender in the multivariable logistic regression models, most FFAs still exhibited elevated concentrations in the LC group (OR > 1, *p* < 0.05), except for C12:1 and the ω6/ω3 PUFAs ratio.

Subgroup analyses were also conducted for both smoking and non-smoking populations (Appendix A), males and females (Appendix A), age not exceeding 50 years old and over 50 years old (Appendix A), as well as populations of normal weight, overweight, and obesity (Appendix A). Similar to results in the overall population, lung cancer was associated with higher levels of most types of FFAs, with odds ratio (OR) values greater than 1. 

### 3.3. Diagnostic Prediction Models

We utilized the metabolic characteristics of the obtained free fatty acids to develop innovative diagnostic methods for lung cancer with machine learning methods (Figure 1). Using the LASSO regression algorithm, we identified 19 fatty acids to distinguish between LC and HC. Subsequently, we constructed 4 logistic regression models in the discovery set, incorporating clinical information, lung tumor biomarkers [24], and the 19 fatty acid features as variables, which were then validated in the test set.

Model 2 incorporated 19 FFAs and clinical features, yielding an area under the receiver operating characteristic curve (AUROC) of 0.830 (95% confidence interval (CI): 0.780–0.880, accuracy: 0.764, precision: 0.712, recall: 0.803, f1 score: 0.755; Figure 1, Table 3). Compared to reference model (AUROC: 0.646 (0.579–0.714), accuracy: 0.620, precision: 0.538, recall: 0.657, f1 score: 0.592) trained with clinical features and tumor biomarkers, the diagnostic performance for lung cancer significantly improved in Model 2 (*p* < 0.001) (Table 4).

### 3.4. Dietary Fatty Acid Intake and Serum FFAs

Dietary fatty acid intakes, including TFAs, SFAs, MUFAs, and PUFAs, were computed from individual FFQs and adjusted for intake per 1000 kcal of energy.

As shown in Figure 2, the correlations between these four types of dietary fatty acids and their corresponding components in serum were analyzed using Spearman correlation analysis. Among them, only SFAs exhibited a notable positive association between dietary intake and serum levels, with a moderate correlation coefficient of 0.358 (*p* < 0.05). TFAs, MUFAs, and PUFAs showed no significant dietary–serum associations, with correlation coefficients of 0.109, 0.116, and 0.022, respectively.

### 3.5. Differential Gene Expression in Fatty Acid Metabolism Pathways

As shown in Figure 3 and Appendix A, genes related to fatty acid metabolism showed significantly distinct expression between lung tumors and normal tissues. Among the 68 relevant genes, 26 genes showed significant upregulation (FDR < 0.05), while 33 genes exhibited significant downregulation, revealing the presence of abnormal fatty acid metabolism in lung cancer tissues.

### 3.6. Correlations between FFAs and Blood Indicators

Spearman correlations were conducted between FFAs and blood indicators in the whole population, HC group, and LC group, respectively, as shown in Appendix A. Several FFAs had positive or negative correlations with TG, TC, LDL-C, HDL-C, ApoB, ApoA1, C-reactive protein, neutrophil counts, neutrophil–leukocyte ratio, lymphocyte counts, and lymphocyte–leukocyte ratio. Moreover, some correlations exhibited differences among different populations. For instance, many FFAs were correlated with lymphocyte counts in the HC group, but lacked these correlations in the LC group, which indicated potential interaction effects between them in lung cancer incidence.

### 3.7. Interaction Effects between FFAs and Immune Factors in Lung Cancer

As shown in Figure 4, in the logistic regression analysis of lung cancer occurrence, some FFAs exhibited significant interaction effects with levels of C-reactive protein and lymphocyte counts. Positive interaction effects indicated mutual enhancement, while negative interaction effects suggested mutual resistance or attenuation. For instance, C20:3 showed a negative interaction effect with C-reactive protein, while C18:3 α had inverse interaction effects with lymphocyte counts.

## 4. Discussion

In this study, an elevated serum free fatty acids pattern was found in lung cancer patients, distinguished from a healthy population. With a substantial sample size of 860 participants, stratified analyses across different demographics were conducted to demonstrate the robustness of the findings. In addition, through joint analysis combining nutritional, tumor-related, and immune-related data, we explored the factors influencing free fatty acids and how they functioned in lung cancer.

Building upon previous research, this study has yielded new points. A study unveiled that the levels of total serum free fatty acids of cancer patients surpassed those in non-cancer patients [13], but the study compared only hospitalized patients’ data, and specific types of fatty acids were not distinguished. Moreover, the performance of the lung cancer diagnostic model constructed based on this approach was not satisfactory (AUROC = 0.545). Another study, focused on the metabolic pathways of C20:4 (arachidonic acid) and C18:2 (linoleic acid) in serum, revealed a 1.8–3.3-fold elevation of their downstream metabolites in lung cancer patients [18]. The general understanding from these studies was that FFAs contributed to the tumor progression by influencing metabolism. In this study, with the LC-MS/MS method employed to quantify common medium- and long-chain fatty acids (C12-C22) in serum, we were able to show that abnormalities in fatty acid levels were not limited to specific compounds. This illustrated a picture wherein a wider variety of fatty acids were linked with lung cancer.

The human body obtains a rich supply of fatty acids from various food sources [25], besides endogenous synthesis. In our population, we observed a lack of correlation between dietary fatty acids and serum free fatty acids, except for SFAs. A meta-analysis of prospective cohort studies revealed that excessive intake of total fat and SFAs posed risk factors for lung cancer, while high intake of PUFAs served as a protective factor [8]. However, we observed elevated serum levels of free fatty acids in lung cancer patients, irrespective of the specific fatty acid types, compared to those in the healthy population. Our results suggested that the elevated concentrations of serum fatty acids might not solely be attributed to an excessive dietary intake of corresponding components. It is worth noting that our dietary data only included a portion of the study cohort, i.e., 95 participants. Thus, further studies are warranted to validate the correlation of serum FFAs and dietary FFA intake.

Fatty acids were essential for cells, participating in cell membrane synthesis, energy metabolism, signaling, and protein acylation [26]. Besides supporting tumorigenesis, lipid-regulated signaling processes played crucial roles in cancer progression and metastasis. In addition to endogenous synthesis, tumors also acquired lipids, including fatty acids, from the extracellular environment. Free fatty acids, due to their accessibility, were preferentially utilized by tumors [27]. RNA-sequencing data supported the notion that abnormal metabolism of fatty acids existed in lung tumor tissues. These differentially expressed genes primarily involved fatty acid beta-oxidation (e.g., ACACB, ACADL, and CPT1A), desaturation of acyl fatty acids (e.g., SCD5), and elongation of carbon chains (e.g., HACD1 and OXSM). These functions exerted effects on cancer cell proliferation, tumor growing, progression, and metastasis [28].

Multiple fatty acids participated in crucial steps promoting tumor progression. Oleic acid (C18:1) could disrupt the dense packing of saturated acyl chains and increase membrane fluidity, which favored the intravasation and formation of tumor cells’ in vivo metastasis [29]. Metabolites of arachidonic acid (C20:4), such as class 20-hydroxyeicosatetraenoic acid, encompassed numerous lipid-signaling mediators that played a central role in cell-signaling cascades relevant to pathological physiology [30]. Recognized as active carcinogens or promoters of tumor growth, elevated expression levels of these metabolites had promoting effects on cancer development. Palmitic acid (C16:1) was absorbed by cancer cells and converted into acetyl-CoA, participating in the downstream transcription factor protein NF-κB, supporting the processes of tumor proliferation and metastasis [31]. Conversely, effectively inhibiting tumors can be achieved by blocking fatty acid metabolism. The application of ND-646, an ACC1 inhibitor, to human NSCLC cell lines nearly eliminated lipid synthesis and accumulation, simultaneously inhibiting cell proliferation and promoting apoptosis [28,32].

For factors associated with lung cancer incidence, some free fatty acids exhibited interactive effects with levels of lymphocytes and C-reactive protein in the bloodstream. These interactions implied that FFAs mighty be involved in the immune response to lung cancer, exerting suppressing effects, which may influence the cancer progression. Tumor cells exhibited distinct metabolic patterns, including lipid metabolism, which altered the local metabolic environment and inhibited anti-tumor immunity in various ways compared to normal stromal cells [33]. Fatty acid metabolism significantly impacted the proliferation and function of cells within the tumor microenvironment [34], involving various cell types that participate in immune functions. The buildup of surplus fatty acids in Natural Killer cells suppressed their cytotoxic activity [35]. Increased uptake of fatty acids by cells, triggered by the elevation of PPARα/δ target genes, hindered the production of IFN-γ and cytotoxic granules. This impairment affected the tumor-associated dendritic cells’ ability to initiate anti-tumor responses by activating T cells [36]. The deletion of FASN and inhibition of lipid synthesis could trigger apoptosis in intra-tumoral Treg cells, leading to a potent anti-tumor response and reduced tumor growth. Existing studies have reported the extensive involvement of fatty acids in tumor immunity. In addition, there were suppressing immune effects of fatty acids on lung cancer. This revealed the presence of complex regulatory mechanisms in the relationship among fatty acids, tumors, and immunity.

Based on existing evidence, we could not simply conclude whether higher FFAs were the cause or consequence of lung cancer incidence. There were mendelian randomization studies that suggested C22:5 ω3, C22:6 ω3, and PUFAs were causes of lung cancer in the European population [27,37,38,39]. However, no such causal relationship has been found between other types of fatty acids and lung cancer yet. On the other hand, it is worth considering whether reducing fatty acid levels could enhance the effectiveness and prognosis of lung cancer treatment. For instance, lipid-lowering statins could help improve the prognosis of lung cancer patients [10]. Perhaps reducing fatty acid levels is how statins achieve their function [40]. These findings suggest promising research and application prospects for the relationship between fatty acids and lung cancer.

The use of blood metabolites for lung cancer diagnosis is garnering increasing attention due to its non-invasive nature and outstanding predictive performance. Lung cancer diagnostic models that screened metabolites from various pathways, including fatty acids, amino acids, uric acid, choline, and purines, reported AUROC values exceeding 0.8 [41,42,43,44,45]. This suggested that these metabolites and their respective pathways might exert a substantial impact on lung cancer development, in which fatty acids make a significant contribution. Furthermore, the application of machine learning methods of artificial intelligence in developing diagnostic models enables the selection of high-dimensional and complex features, addresses nonlinear relationships, and achieves higher predictive performance, which is suitable for models constructed with multiple metabolites [46]. In this study, using free fatty acids as candidate predictors, the AUROC reached 0.830, significantly outperforming the predictive efficacy compared to models incorporating clinical features and traditional tumor markers. This demonstrates superior clinical applicability compared to some tumor markers in lung cancer diagnosis. While it did not reach the diagnostic level of some broad-spectrum metabolite models (AUROC > 0.9), the broad-spectrum mass spectrometry testing is non-specific and generally used for preliminary screening, with limited application in quantitative analysis of specific analytes. While the FFA-based model performed well, further validation against broader diagnostic models is needed in future studies. The free fatty acid LC-MS/MS quantification method discovered in this study partially overcame these challenges, showing great potential for further research and widespread application in lung cancer diagnosis.

## 5. Strengths and Limitations

This study had several strengths. First, we established a validated and reliable liquid chromatography–mass spectrometry (LC-MS/MS) method to measure 27 common free fatty acids in the human body. Second, the study population consisted of 430 lung cancer patients and 430 healthy controls, providing a large sample size that facilitated subgroup analysis and could serve as a reference for the clinical application of free fatty acids. Third, for the first time, we integrated multi-omics approaches, including nutritional, gene expression, and immune indicator data, to explore the influencing factors and effects associated with free fatty acids.

This study also had some limitations. As a case-control study, it could not establish a causal relationship between serum levels of free fatty acids and the occurrence of lung cancer. The dietary data were limited to 95 participants, which may not be representative of the entire cohort.

## 6. Conclusions

Lung cancer patients had elevated levels of serum free fatty acids compared to healthy individuals, and the elevation appeared not to be substance specific. This might not be attributed to excessive intake of relevant fatty acids from the diet, but rather a pathological phenomenon associated with lung cancer. Abnormal expression of fatty acid metabolic pathway genes was observed, and serum free fatty acids might also exert complex effects on tumor immunity. These findings will complement research on fatty acid metabolism in lung cancer and provide insights into potential intervention targets.

## Figures and Tables

**Figure 1 nutrients-16-02884-f001:**
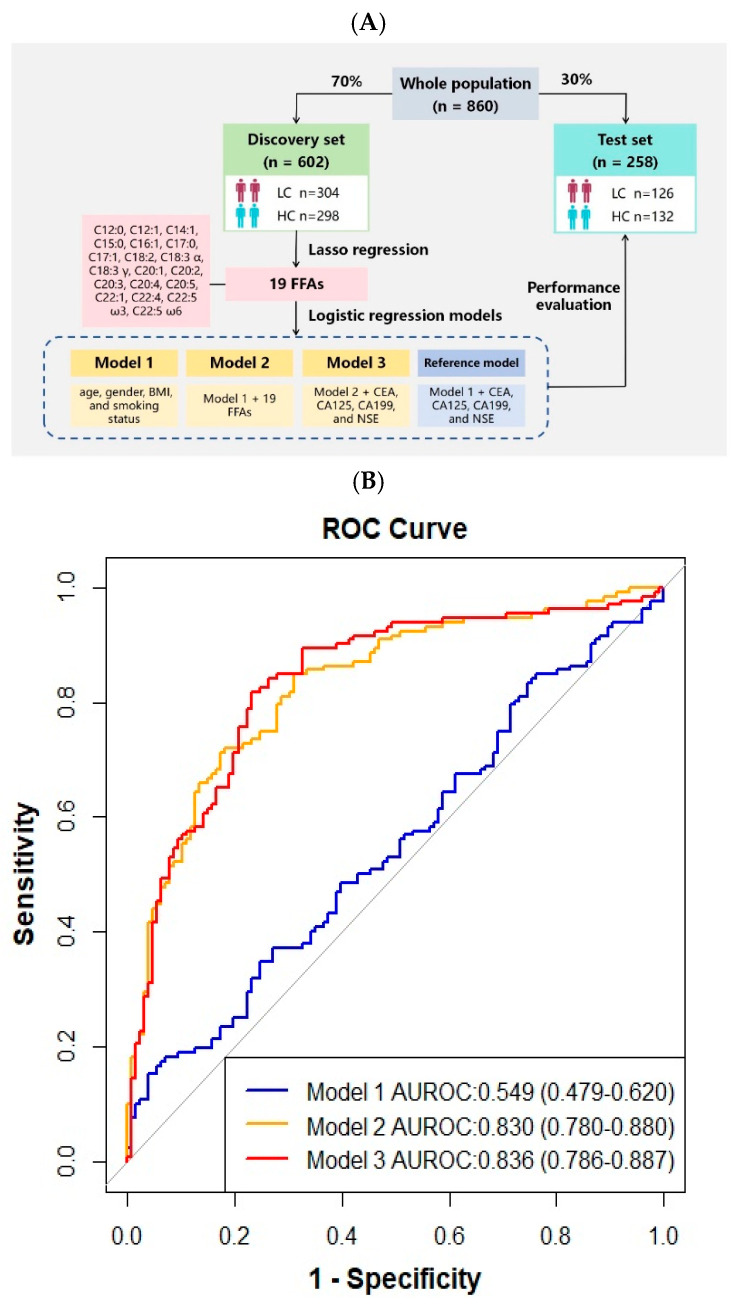
Machine learning-derived prediction models for LC diagnosis. (**A**) Design of the modeling workflow. LASSO regression and logistic regression were adopted for feature selection and model training. The models were validated in the test set. (**B**) The receiver operating characteristic (ROC) curve for the diagnosis of LC patients in the test set.

**Figure 2 nutrients-16-02884-f002:**
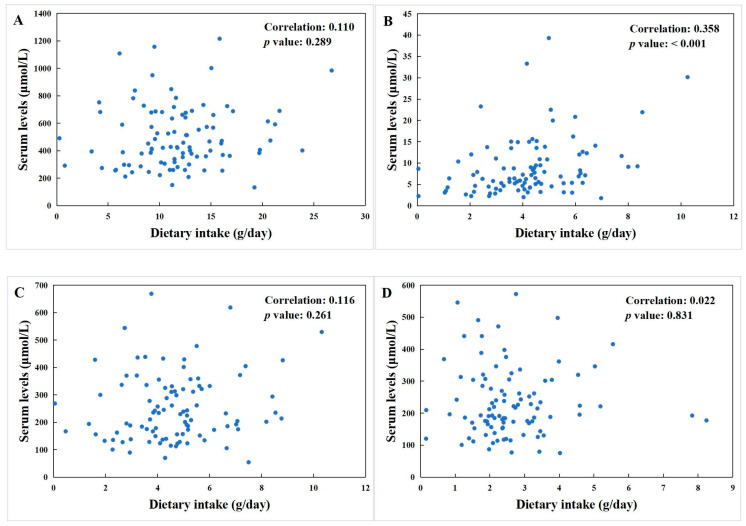
Spearman correlation between dietary intake and serum levels of (**A**) total fatty acids (TFAs), (**B**) saturated fatty acids (SFAs), (**C**) monounsaturated fatty acids (MUFAs), and (**D**) polyunsaturated fatty acids (PUFAs).

**Figure 3 nutrients-16-02884-f003:**
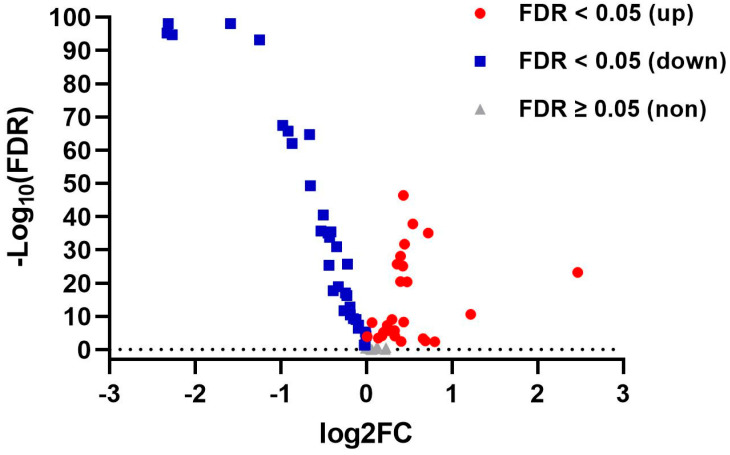
Volcano plot of lipid metabolism genes’ RNA-seq of tumors (*n* = 346) vs. normal tissues (*n* = 401). FDR, false discovery rate.

**Figure 4 nutrients-16-02884-f004:**
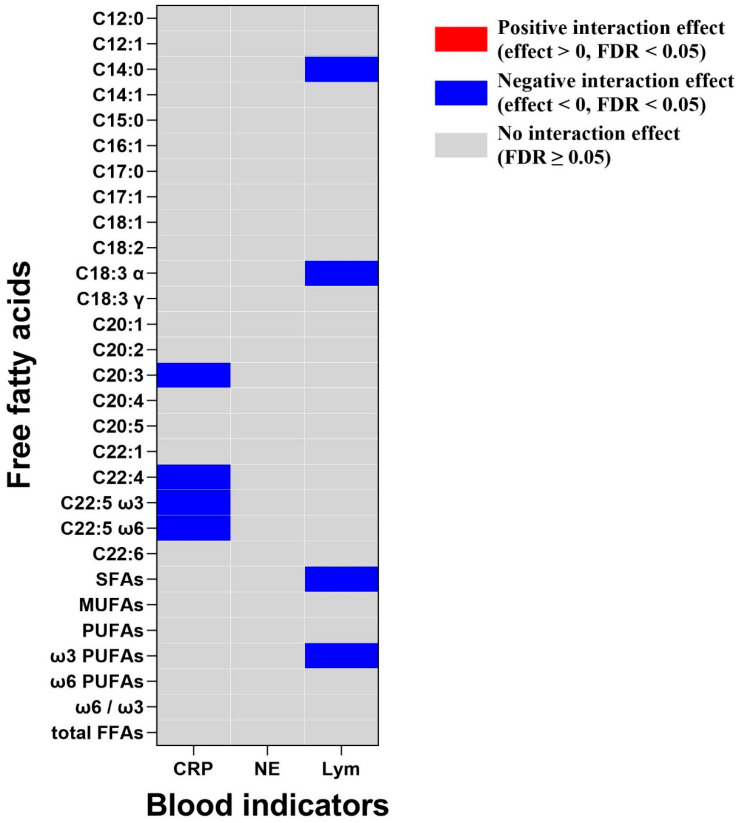
Interaction effects between FFAs and blood parameters in multivariate logistic regression models. Models were adjusted for age, gender, BMI, smoking status, and diabetes status. Abbreviations: CRP, C-reactive protein; NE, neutrophil counts; Lym, lymphocyte counts.

**Table 1 nutrients-16-02884-t001:** Baseline characteristics of the study population.

	Participants			*p*-Value
	Total (*n* = 860)	Healthy Controls (*n* = 430)	Lung Cancer Patients (*n* = 430)	
Gender, (%)				
Male	378 (44.0)	189 (44.0)	189 (44.0)	1
Female	482 (56.0)	241 (56.0)	241 (56.0)	
Age (years)	48.03 [37.00, 58.85]	48.00 [37.00, 58.00]	48.84 [36.90, 59.24]	0.344
Body mass index (kg/m^2^)	23.42 [21.12, 25.59]	23.63 [21.22, 25.72]	23.12 [21.05, 25.37]	0.075
Smoking status, (%)				
Smoker	224 (26.0)	101 (23.5)	123 (28.6)	0.017
Non-smoker	622 (72.3)	326 (75.8)	296 (68.8)	
Unknown	14 (1.6)	3 (0.7)	11 (2.6)	
Diabetes, (%)	38 (4.4)	16 (3.7)	22 (5.1)	0.512
Blood indicators				
Total cholesterol (mmol/L)	4.95 [4.30, 5.66]	4.95 [4.28, 5.61]	4.94 [4.32, 5.71]	0.548
Triglycerides (mmol/L)	1.15 [0.82, 1.76]	1.10 [0.81, 1.70]	1.17 [0.83, 1.85]	0.238
LDL cholesterol (mmol/L)	2.67 [2.18, 3.27]	2.70 [2.20, 3.26]	2.63 [2.16, 3.30]	0.735
HDL cholesterol (mmol/L)	1.34 [1.11, 1.58]	1.31 [1.07, 1.57]	1.35 [1.14, 1.59]	0.088
Apolipoprotein A1 (g/L)	1.39 [1.23, 1.56]	1.39 [1.22, 1.56]	1.38 [1.25, 1.56]	0.944
Apolipoprotein B (g/L)	0.90 [0.72, 1.08]	0.90 [0.72, 1.05]	0.90 [0.73, 1.11]	0.264
C-reactive protein (mmol/L)	1.10 [0.50, 2.20]	0.80 [0.40, 1.50]	1.55 [0.90, 3.20]	<0.001
Neutrophil counts (×10^9^/L)	3.58 [2.84, 4.50]	3.24 [2.67, 3.88]	4.03 [3.18, 5.59]	<0.001
Lymphocyte counts (×10^9^/L)	1.77 [1.44, 2.17]	1.85 [1.55, 2.30]	1.63 [1.32, 2.07]	<0.001
Histopathology				
Histology, (%)				
Adenocarcinoma			406 (94.4)	
Other types			21 (4.9)	
Unknown			3 (0.7)	
Stage, (%)				
In situ carcinoma			65 (15.1)	
Early stage			349 (81.2)	
Advanced stage			16 (3.7)	

Abbreviations: LDL, low-density lipoprotein; HDL, high-density lipoprotein. Data are presented as a number (percentage) for categorical variables and median [Q1, Q3] for continuous variables.

**Table 2 nutrients-16-02884-t002:** Comparison of serum free fatty acids (FFAs) between lung cancer patients and healthy controls.

FFAs	FFA Concentrations (μmol/L)		Comparison between Groups		Multivariate Logistic Regression Model	
	Healthy Controls (*n* = 430)	Lung Cancer Patients (*n* = 430)	Fold Change	*p*-Value	OR (95% CI)	*p*-Value
C12:0	0.97 [0.34, 2.14]	1.25 [0.57, 2.56]	1.29	0.004	1.107 (1.038–1.187)	0.003
C12:1	0.30 [0.06, 0.69]	0.51 [0.11, 1.01]	1.69	<0.001	1.137 (0.995–1.368)	0.128
C14:0	5.15 [3.29, 7.71]	7.15 [4.38, 10.56]	1.39	<0.001	1.129 (1.091–1.170)	<0.001
C14:1	0.31 [0.07, 0.72]	0.51 [0.18, 0.97]	1.62	<0.001	2.018 (1.567–2.640)	<0.001
C15:0	0.61 [0.32, 1.00]	0.74 [0.37, 1.24]	1.22	0.001	1.428 (1.168–1.764)	0.001
C15:1		Below detection limit				
C16:1	16.41 [11.06, 25.22]	24.90 [15.12, 39.56]	1.52	<0.001	1.048 (1.036–1.059)	<0.001
C17:0	1.19 [0.80, 1.64]	1.58 [0.99, 2.32]	1.33	<0.001	1.648 (1.409–1.942)	<0.001
C17:1	1.07 [0.69, 1.55]	1.47 [0.81, 2.22]	1.37	<0.001	1.742 (1.481–2.066)	<0.001
C18:1	184.52 [129.25, 262.41]	254.32 [165.60, 367.92]	1.38	<0.001	1.005 (1.004–1.006)	<0.001
C18:1 T		Below detection limit				
C18:2	170.02 [118.33, 239.16]	247.83 [159.33, 367.25]	1.46	<0.001	1.005 (1.004–1.007)	<0.001
C18:2 T		Below detection limit				
C18:3 α	9.51 [6.99, 13.20]	14.77 [9.35, 20.88]	1.55	<0.001	1.111 (1.086–1.137)	<0.001
C18:3 γ	1.09 [0.77, 1.50]	1.64 [1.02, 2.38]	1.50	<0.001	2.579 (2.108–3.194)	<0.001
C20:0		Below detection limit				
C20:1	1.83 [1.24, 2.57]	2.82 [1.70, 4.46]	1.54	<0.001	1.516 (1.378–1.678)	<0.001
C20:2	1.85 [1.29, 2.50]	2.65 [1.66, 3.63]	1.43	<0.001	1.794 (1.578–2.054)	<0.001
C20:3	1.21 [0.85, 1.59]	1.79 [1.12, 2.35]	1.48	<0.001	2.615 (2.134–3.243)	<0.001
C20:4	5.54 [4.42, 7.33]	6.33 [4.73, 8.62]	1.14	<0.001	1.138 (1.079–1.202)	<0.001
C20:5	0.43 [0.27, 0.65]	0.54 [0.33, 0.80]	1.25	<0.001	2.142 (1.497–3.118)	<0.001
C22:0		Below detection limit				
C22:1	0.24 [0.15, 0.39]	0.36 [0.18, 0.68]	1.51	<0.001	3.366 (2.291–5.194)	<0.001
C22:4	0.86 [0.60, 1.22]	1.26 [0.79, 1.78]	1.45	<0.001	2.972 (2.323–3.855)	<0.001
C22:5 ω3	0.85 [0.59, 1.25]	1.21 [0.76, 1.72]	1.42	<0.001	2.234 (1.787–2.825)	<0.001
C22:5 ω6	0.47 [0.33, 0.71]	0.67 [0.43, 0.95]	1.42	<0.001	4.503 (2.995–6.918)	<0.001
C22:6 ω3	5.23 [3.95, 7.08]	6.33 [4.38, 8.71]	1.21	<0.001	1.133 (1.084–1.187)	<0.001
SFAs	6.17 [3.83, 9.93]	8.65 [5.22, 13.09]	1.40	<0.001	1.079 (1.054–1.107)	<0.001
MUFAs	203.78 [145.66, 293.28]	288.23 [186.34, 406.18]	1.41	<0.001	1.005 (1.004–1.006)	<0.001
PUFAs	198.35 [138.56, 274.79]	283.38 [185.48, 413.06]	1.43	<0.001	1.005 (1.004–1.006)	<0.001
ω3 PUFAs	16.49 [12.60, 22.44]	23.88 [15.68, 31.79]	1.45	<0.001	1.072 (1.055–1.089)	<0.001
ω6 PUFAs	176.02 [123.41, 245.75]	253.39 [164.56, 374.26]	1.44	<0.001	1.005 (1.004–1.007)	<0.001
ω6/ω3 PUFAs	10.40 [8.57, 12.52]	11.05 [8.91, 13.70]	1.06	0.022	1.013 (0.988–1.042)	0.324
Total FFAs	422.63 [294.03, 588.74]	600.75 [385.44, 836.85]	1.42	<0.001	1.003 (1.002–1.003)	<0.001

Abbreviations: SFAs, saturated fatty acids; MUFAs, monounsaturated fatty acids; PUFAs, polyunsaturated fatty acids; ω3 PUFAs, ω3 polyunsaturated fatty acids; ω6 PUFAs, ω6 polyunsaturated fatty acids; ω6/ω3 PUFAs, ratio of ω6 PUFAs to ω3 PUFAs. Data are presented as median [Q1, Q3] for data with a skewed distribution. Fold change calculation: median value of LC group/median value of HC group. Multivariate logistic regression models were adjusted for age, gender, BMI, smoking status, and diabetes status.

**Table 3 nutrients-16-02884-t003:** Evaluation of machine learning-derived prediction models for LC diagnosis in the test set.

	Variables in Model	AUROC (95% CI)	*p*-Value	Accuracy	Precision	Recall	F1 Score
Model 1	Age, gender, BMI, and smoking status	0.549 (0.479–0.620)	0.009	0.535	0.470	0.554	0.509
Model 2	Model 1 + 19 FFAs	0.830 (0.780–0.880)	<0.001	0.764	0.712	0.803	0.755
Model 3	Model 2 + CEA, CA125, CA199, and NSE	0.836 (0.786–0.887)	<0.001	0.760	0.727	0.787	0.756
Reference model	Model 1 + CEA, CA125, CA199, and NSE	0.646 (0.579–0.714)	Reference	0.620	0.538	0.657	0.592

Accuracy = (TP + TN)/(TP + FP + TN + FN); Precision = TP/(TP + FP); Recall = TP/(TP + FN); F1 Score = 2 × Precision × Recall/(Precision + Recall). Abbreviations: AUROC, area under the receiver operating characteristic curve; BMI, body mass index; CEA, carcinoembryonic antigen; CA125, carbohydrate antigen 125; CA199, carbohydrate antigen 199; NSE, neuron-specific enolase; TP, true positive; TN, true negative; FP, false positive; FN, false negative.

**Table 4 nutrients-16-02884-t004:** Multivariate logistic regression results and parameters for Model 2.

Variables	Coefficient	*p*-Value	OR (95% CI)	Value Assigned in Model
Constant	−0.578	0.532		
Age	−0.002	0.808	0.998 (0.982–1.014)	
Gender	0.101	0.709	1.106 (0.652–1.888)	Male: 0; Female: 1
BMI	−0.058	0.069	0.944 (0.886–1.004)	
Smoking status	0.655	0.020	1.926 (1.111–3.368)	Smoker: 1; Non-smoker: 0
C12:0	0.084	0.096	1.088 (0.988–1.208)	
C12:1	−0.079	0.521	0.924 (0.654–1.078)	
C14:1	−0.383	0.314	0.682 (0.323–1.441)	
C15:0	−0.431	0.153	0.650 (0.351–1.148)	
C16:1	0.066	<0.001	1.068 (1.032–1.107)	
C17:0	0.439	0.106	1.551 (0.925–2.700)	
C17:1	−1.471	<0.001	0.230 (0.112–0.451)	
C18:2	0.001	0.576	1.001 (0.999–1.003)	
C18:3 α	−0.005	0.857	0.995 (0.944–1.053)	
C18:3 γ	0.610	0.015	1.840 (1.138–3.043)	
C20:1	0.025	0.807	1.026 (0.838–1.263)	
C20:2	−0.038	0.855	0.962 (0.640–1.477)	
C20:3	1.545	<0.001	4.688 (2.168–10.565)	
C20:4	−0.190	0.001	0.827 (0.737–0.926)	
C20:5	0.998	0.029	2.714 (1.128–6.807)	
C22:1	0.666	0.006	1.947 (1.284–3.316)	
C22:4	0.335	0.472	1.398 (0.561–3.511)	
C22:5 ω3	−1.582	<0.001	0.206 (0.083–0.484)	
C22:5 ω6	0.789	0.152	2.202 (0.765–6.596)	

## Data Availability

The individual-level data for this study have been deposited on the Zhejiang University School of Public Health Computational Platform. Collaborators interested in accessing the individual-level data should contact the corresponding author. Access requests will be reviewed by the project committee.

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
