# Peer review of "Non-Specific Elevated Serum Free Fatty Acids in Lung Cancer Patients: Nutritional or Pathological?"

_nutrients, 2024, doi:10.3390/nu16172884_

Round 1

Reviewer 1 Report

Comments and Suggestions for Authors

The authors present interesting data about "Non-specific elevated serum free fatty acids in lung cancer patients: nutritional or pathological". Authors noted that association between fatty acid levels and related factors including nutritional intake, tumor metabolism, and  tumor immunity is rarely discussed.

The manuscript is well written, has important clinical message, and should be of great interest to the readers. The results are well presented.

However, several premises must be discussed:

  1. The results: What is the analysis of FFA only? The methodology seems to be well described. I wonder what the FFA percentage would be and whether such amounts could actually be detected in serum if the procedure were changed: extract whole lipids and on that basis determine what % are FFA and then identify them. What do the authors think about this.
  2. Discussion: line 315-325 “the lack of lack of correlation between dietary fatty acids and serum free fatty acids, except for SFAs”. Does this mean that all other FAs come from endogenous synthesis? And 18:2n6 and 18:3n3, which are essential FAs because they cannot be synthesized de novo?
  3. Line 386 – “lipid-lowering statins could help improve the prognosis of lung cancer patients”. It is not certain that statins reduce free fatty acid levels or maybe the proportion of FAs in complex lipids.
  4. Discussion: The authors did not consider lipolysis of triglycerides and release of free acids from cells as sources in serum. What about lipotoxicity? What were the specific BMI values of the patients? It is also worth splitting and reviewing patients by BMI:  Multivariate logistic regression analyses of serum free fatty acids (FFAs) between lung cancer patients and healthy controls stratified by BMI.

Minor comments:

  1. Line 122 – should be 22:6n-3 – add to the whole text;
  2. Line 139 Table S8 – the first should be a Table 1, change the numbering order;
  3. How many patients complete the FFQ?
  4. Line 312 – long chain FA start at C13; C12 is still medium chain FA;
  5. Line 355 - which one is 22:5? Because there is 22:5n3 and 22:5n6;
  6. Lines 356-358 the missing reference;
  7. Lines 381-383 repeated sentences from the page above.

Author Response

Comment 1. The results: What is the analysis of FFA only? The methodology seems to be well described. I wonder what the FFA percentage would be and whether such amounts could actually be detected in serum if the procedure were changed: extract whole lipids and on that basis determine what % are FFA and then identify them. What do the authors think about this.

Response: We thank the reviewer for the comment. Each FFA was quantified using an absolute quantification method rather than an estimation based on overall proportions. For each FFA to be detected, we first used a standard solution with a known concentration and obtained the mass spectrometry response values at different concentrations in the LC-MS/MS. We then constructed the corresponding standard curves, and the R² for all the FFAs were > 0.99. Thus, when measuring the concentration of FFAs in blood samples, we calculated the absolute concentration of each FFA by comparing its mass spectrometry response to the standard curve. Therefore, the concentrations of the 27 FFAs were independently measured.

We used rigorous method to control the extraction of FFAs. On one hand, since the standard substances were also dissolved in serum, we assumed that the fatty acids in the human serum and the fatty acid standards could be compared based on the same extraction efficiency when the exact same extraction method was applied. On the other hand, we employed a relatively exhaustive fatty acid extraction method (i.e., methanol as solvent, shaken at 1450 rpm, 25℃ for 10 min) to extract as much fatty acid as possible. This approach is commonly used in fatty acid metabolomics. This procedure was rigorously followed during the processing of large serum samples to ensure comparability between samples.

Regarding the broader application of this detection method, when implementing this approach on another LC-MS/MS system, it is necessary to first analyze standards of known concentrations to obtain the concentration-response standard curve, as done in this study. Only after this step can the blood samples be analyzed to achieve absolute quantification of fatty acids.

We added the following text on Page 12 in the supplementary materials to provide more details.

“For each FFA to be detected, we first used a standard solution with a known concentration and obtained the mass spectrometry response values at different concentrations in the LC-MS/MS. We then constructed the corresponding standard curves with R² > 0.99. Thus, when measuring the concentration of FFAs in serum samples, we calculated the absolute concentration of each compound by comparing its mass spectrometry response to the standard curve. Therefore, the concentrations of the 27 FFAs were independently measured.”

Comments 2. Discussion: line 315-325 “the lack of lack of correlation between dietary fatty acids and serum free fatty acids, except for SFAs”. Does this mean that all other FAs come from endogenous synthesis? And 18:2n6 and 18:3n3, which are essential FAs because they cannot be synthesized de novo?

Response: We thank the reviewer for the comment, and we are sorry if our description led to confusion. We did not mean that all other FAs come from endogenous synthesis. The FFAs in serum originated from multiple sources, including dietary intake, endogenous de novo synthesis, interconversion among fatty acids, release of free acids from other cells, and the lipolysis of triglycerides. For example, 18:2n6 and 18:3n3 could not be synthesized de novo, but their free fatty acid forms could be supplemented through dietary sources or the lipolysis of triglycerides.

Comments: 3. Line 386 – “lipid-lowering statins could help improve the prognosis of lung cancer patients”. It is not certain that statins reduce free fatty acid levels or maybe the proportion of FAs in complex lipids.

Response: We thank the reviewer for the comment. As you indicated by the reviewer, we recognized that this statement might be controversial, as it was based on evidence from two different studies.

We revised the following text on Page 12 Lines 382-383 in the Discussion section to clarify.

Comments: 4. Discussion: The authors did not consider lipolysis of triglycerides and release of free acids from cells as sources in serum. What about lipotoxicity? What were the specific BMI values of the patients? It is also worth splitting and reviewing patients by BMI:  Multivariate logistic regression analyses of serum free fatty acids (FFAs) between lung cancer patients and healthy controls stratified by BMI.

Response: We thank the reviewer for the comment. The FFAs in serum originated from multiple sources, including dietary intake, endogenous de novo synthesis, interconversion among fatty acids, release of free acids from other cells, and the lipolysis of triglycerides. The FFAs measured in our study include FFAs from all sources.

In our study, we have accounted for the effects of diabetes, obesity, and triglycerides, all of which are closely related to lipotoxicity [1]. Specifically, we adjusted for diabetes and obesity as covariates in our logistic regression models. Additionally, there was no significant difference in triglyceride levels between the case and control groups, indicating that triglyceride levels have been controlled for in our analysis. Therefore, from an epidemiological perspective, we supposed that the impact of lipotoxicity had been considered.

The distribution of the BMI values are provided in Table 1 for the overall participants, the controls, and the cases. Also, as suggested by the reviewer, we added an additional analysis stratified by BMI, and the results are shown in Table S5. Multivariate logistic regression analyses of serum free fatty acids (FFAs) between lung cancer patients and healthy controls stratified by BMI in Supplemental Materials. In subgroups of normal weight (BMI < 24), overweight (24 ≤ BMI < 28), and obesity (BMI ≥ 28), lung cancer had higher levels of most types of FFAs, with odds ratio (OR) values greater than 1.

We added the following text on Page 5 Lines 212-213 in the Results section to describe the results.

“as well as populations of normal weight, overweight, and obesity (Table S5)”

Minor comments:

Comments 1. Line 122 – should be 22:6n-3 – add to the whole text;

Response: We thank the reviewer for catching this typo. We revised it as suggested in the resubmission.

Comments 2. Line 139 Table S8 – the first should be a Table 1, change the numbering order.

Response: We thank the reviewer for catching this typo. We revised it as suggested in the resubmission.

Comments 3. How many patients complete the FFQ?

Response: Of the 95 participants, 5 patients complete the FFQ. We provided additional clarification on this issue and included it as a limitation on Page 3 Lines 137-138 as shown below.

“(90 healthy controls and 5 patients)”

Comments 4. Line 312 – long chain FA start at C13; C12 is still medium chain FA;

Response: We revised the wording to "medium and long chain" on Page 11 Line 314.

Comments 5. Line 355 - which one is 22:5? Because there is 22:5n3 and 22:5n6;

Response: It should be 22:5n3, and we revised this on Page 12 Line 377.

Comments 6. Lines 356-358 the missing reference;

Response: The missing references were added (Page 12, Line 378).

Comments 7. Lines 381-383 repeated sentences from the page above.

Response: We thank the reviewer for catching this, and the repeated sentences were deleted in the revised manuscript.

Reviewer 2 Report

Comments and Suggestions for Authors

1. Although the experiments presented in the manuscript are well-executed, there is existing research examining the link between elevated serum free fatty acids (FFAs) and lung cancer. there is one study specifically evaluated the diagnostic potential of FFAs as biomarkers for lung cancer (https://link.springer.com/article/10.1007/s00432-019-03095-8).

3. The conclusion regarding the lack of correlation between dietary intake and serum FFAs seams to be valid/ However, the small sample size of the dietary study (95 participants) could be a a limitation.

4. In my opinion it is important to clarify that while the FFA-based model performs well, it may still need validation against broader diagnostic models.

5. The dietary data was limited to only 95 participants. This may  not be representative of the entire cohort.

6. Nevertheless, presented study shows elevated serum FFAs in lung cancer patients it does not establish a clear causal relationship between elevated FFAs and lung cancer.

7. The study highlighted interactions between FFAs and immune responses, but it could beenfit more from the disccusion of the complex interplay between fatty acids, tumor immunity, and cancer progression.

Comments on the Quality of English Language

It requires only some small language editing

Author Response

Comments: 1. Although the experiments presented in the manuscript are well-executed, there is existing research examining the link between elevated serum free fatty acids (FFAs) and lung cancer. there is one study specifically evaluated the diagnostic potential of FFAs as biomarkers for lung cancer (https://link.springer.com/article/10.1007/s00432-019-03095-8).

Response: We thank the reviewer for the comment. This paper (https://link.springer.com/article/10.1007/s00432-019-03095-8) constructed a diagnostic model based on the amount of total FFAs, while our study built a model using individual FFAs. Additionally, the diagnostic performance of our model surpassed that of the total FFAs model. We added the following text on Page 11 Lines 308-310 in the Discussion section to discuss this study.

“Moreover, the performance of the lung cancer diagnostic model constructed based on this approach was not satisfactory (AUROC = 0.545).”

Comments 3. The conclusion regarding the lack of correlation between dietary intake and serum FFAs seams to be valid/ However, the small sample size of the dietary study (95 participants) could be a a limitation.

Response: We thank the reviewer for the comment. We added the following text on Page 13 Lines 417-418 in the Strengths and limitations section to list this as a study limitation.

Comments 4. In my opinion it is important to clarify that while the FFA-based model performs well, it may still need validation against broader diagnostic models.

Response: In this study, we compared our model with diagnostic models constructed using clinical features and tumor markers. As indicated by the reviewer, we added the following text on Page 12 Lines 401-403 to further clarify.

“While the FFA-based model performs well, further validation against broader diagnostic models is needed in future studies.”

Comments 5. The dietary data was limited to only 95 participants. This may not be representative of the entire cohort.

Revisions: We included it as an additional limitation (Page 13, Line 417-418).

Comments 6. Nevertheless, presented study shows elevated serum FFAs in lung cancer patients it does not establish a clear causal relationship between elevated FFAs and lung cancer.

Response: It is the limitation of our study (Page 13, Line 415-417).

Comments 7. The study highlighted interactions between FFAs and immune responses, but it could beenfit more from the disccusion of the complex interplay between fatty acids, tumor immunity, and cancer progression.

Response: Thank you for your insight comment. The study highlighted interactions between FFAs and immune responses, a deeper discussion on the complex interplay between fatty acids, tumor immunity, and cancer progression could indeed provide more insights. We have added discussion in revised manuscript (Line 360, 369-371). However, since most lung cancer patients in this study were newly diagnosed and had not yet reached the 3- or 5-year follow-up periods needed to gather data on cancer progression or outcomes, a detailed analysis of these aspects was challenging. We plan to address these topics in future research to better understand their implications for cancer progression and immunity.